# Federated Learning for Edge Computing: A Survey

Alexander Brecko [1,*] , Erik Kajati [1,*] , Jiri Koziorek [2] and Iveta Zolotova [1]

1   Department of Cybernetics and Artificial Intelligence, Faculty of Electrical Engineering and Informatics, Technical University of Kosice, 042 00 Kosice, Slovakia
2   Department of Cybernetics and Biomedical Engineering, Faculty of Electrical Engineering and Computer Science, VSB—Technical University of Ostrava, 708 00 Ostrava, Czech Republic
*   Correspondence: alexander.brecko@tuke.sk (A.B.); erik.kajati@tuke.sk (E.K.)

**Abstract:** New technologies bring opportunities to deploy AI and machine learning to the edge of the network, allowing edge devices to train simple models that can then be deployed in practice. Federated learning (FL) is a distributed machine learning technique to create a global model by learning from multiple decentralized edge clients. Although FL methods offer several advantages, including scalability and data privacy, they also introduce some risks and drawbacks in terms of computational complexity in the case of heterogeneous devices. Internet of Things (IoT) devices may have limited computing resources, poorer connection quality, or may use different operating systems. This paper provides an overview of the methods used in FL with a focus on edge devices with limited computational resources. This paper also presents FL frameworks that are currently popular and that provide communication between clients and servers. In this context, various topics are described, which include contributions and trends in the literature. This includes basic models and designs of system architecture, possibilities of application in practice, privacy and security, and resource management. Challenges related to the computational requirements of edge devices such as hardware heterogeneity, communication overload or limited resources of devices are discussed.

**Keywords:** federated learning; artificial intelligence; machine learning; applications of FL; frameworks of FL; FL on edge devices; communications

## 1. Introduction

The proliferation of smart devices, mobile networks and computing has ushered in a new era of the Internet of Things (IoT), which is poised to significantly advance all aspects of our modern lives, including a smart healthcare system, smart transportation infrastructure, smart cities infrastructure, etc. The IoT, connecting a wide range of devices to the Internet while enabling sensing and computing, has grown rapidly in recent years. Nowadays, computing technology and services can be observed everywhere in the information-centric era. Most of them are moving from cloud services and platforms to edge services and devices [1]. These devices generate a large amount of data that needs to be processed. Through the IoT, which connects a wide range of smart devices, we can access enormous amounts of user data that will give us insights, enable us to build machine learning models that are specifically suited for given tasks, and ultimately enable us to provide high-quality intelligent services and products. With the arrival of the edge computing paradigm, a large amount of computation and data processing moves from the cloud to the edge of the network. Edge computing is primarily concerned with transmitting data among devices at the edge, closer to where user applications are located [2], rather than to a centralized server. In consideration of the performance of these devices, not only calculations and data processing, but also simple tasks of artificial intelligence (AI) and machine learning (ML) can be moved from the cloud to the edge. For companies not disposing with the capacity to create separate data centres for their operations, this new computing paradigm offers

significant cost reductions. Instead, engineers can create a dependable network of smaller, less expensive edge devices.

AI and ML are being used more and more frequently in today's world and are given more emphasis [3]. Significant advances in various areas, including speech, image, text, and translation [4] were made. For learning purposes, traditional ML requires collecting large amounts of data. While there is no denying that ML is on the rise, there might be severe privacy concerns. The biggest problem is that millions of images, videos, or data are collected centrally on a single server. The problem is that there can be much private information in photographs and videos, including faces, car number plates, computer screens, and the sounds of conversations. If this server is attacked, a large amount of sensitive data could be leaked that could be misused. Corporations should have such data well protected. One possibility is to not have everything stored on one server but to distribute it to multiple servers.

Recently, it is possible to encounter AI and ML methods more often. They are used for training predictions or decision models on the network's edge without sending and processing everything to backend data centers and clouds. In addition to these concepts, it is possible to encounter the term federated learning (FL). FL is an approach that started to come into awareness in 2016 [5], when Google introduced this term. That is why FL has been one of the most important research topics. Many researchers worldwide have been working on it since. FL is widely acknowledged as a foundational tool for next-generation AI. It is a complex method for handling distributed data training. It enables the cooperative training of high-quality joint models by combining and averaging locally calculated updates submitted by IoT devices. It has become a trendy topic nowadays, which is discussed in ML and collaboration of multiple devices simultaneously, which are involved in training for privacy [6,7]. Federated learning allows the training of new models on multiple devices simultaneously without the need to have the data stored in a central data center. FL has been used in many applications, including medical, Industry 5.0 and IoT, mobile applications, or transportation [8].

## 2. Literature Review

According to our surveys, the possibilities of FL and EC are being discussed more often. Until now, most of these studies dealt with these areas independently. In addition, most of the literature does not consider the challenges related to hardware requirements and the most modern technologies, such as ARM architectures, which are starting to dominate all edge devices. According to our findings, microcomputers with ARM architecture have very poor support for deploying AI and ML at the network's edge, and existing frameworks do not always support this type of architecture.

The authors of [9] analyzed edge intelligence specifically for 6G networks and also summarized the benefits of this technology. In addition, Cui et al. in [10] reviewed ML applications for IoT control and presented an overview of approaches to offload computing capacity in mobile edge computing. Abbas et al. [11] reviewed MEC architecture and applications. In addition, in [12], the authors review computational offloading modeling for EC, the technology and methods used, and discuss communication, computational, and energy harvesting models. However, some articles deal with FL application review in general, a comprehensive review in the field of communication and network, or articles that deal with FL optimization. Ramu et al. [13] focused on an overview of FL and the digital twin for smart cities, in which they describe concepts, benefits, and future directions. The authors of [14] focused on the future directions and possibilities of providing FL. In the [15] survey, the authors mainly present and analyze ideas based on differential privacy to guarantee user privacy during deep federated learning. Xuezhen Tu et al. [16]. In this paper, I provide a comprehensive overview of economic and game theoretic approaches proposed in the literature to design various schemes to motivate data owners to participate in the FL training process. This study [17] explores FL in-depth, focusing on application and system platforms, mechanisms, real-world applications, and process contexts. However, there is also research

such as [18–20] or [21], which focus on the integration of blockchain technologies associated with edge computing applications. The problem is that blockchain, EC, FL, and hardware requirements for devices or hardware on which these technologies can work without restrictions are not included in all cases. In the Table 1, we summarized an overview of other articles from which we studied information because they focused on FL, EC, or future directions (FD). We also focused on articles describing the possibilities of deploying AI, ML, and FL on the ARM architecture of microcomputers, but we could not find such articles. However, it was possible to find articles that describe, at least to some extent, the EC, FL, and hardware (HW) options that are currently available. Abdel Wahab col. [22] provides a comprehensive guide to federated education and related concepts, technologies, and approaches to education. They explore the applications and future directions of federated learning in communication and networking. They also partially describe the possibilities of deploying AI at the network's edge, where they list edge devices on which these calculations for AI can be performed. However, they do not indicate the possibilities of deploying FL on these devices but focus on future directions and challenges related to FL and EC.

The article focuses on a systematic overview of the technology and methods currently used in FL and EC. It is also important to provide an overview of the hardware limitations that may arise with EC from computing capacity, communication, privacy, and heterogeneity of data and devices that can be used. The first part describes the essential characteristics of FL, the structure of FL, the architectures of FL, and methods of communication. In the second part, the article focuses on an overview of frameworks that can be used with FL. This part focused primarily on obtaining information and supporting ARM architectures and microcomputers. In the third part, the article analyzes all the possibilities of the FL application. The last part focuses on devices with limited computing and communication resources and challenges that need to be solved with EC and FL.

**Table 1.** Comparison of surveys.

| Source | Year | Description | EC | FL | HW | FD |
|---|---|---|---|---|---|---|
| [23] | 2017 | A Survey on Mobile Edge Networks: Convergence of Computing, Caching and Communications | • | | | • |
| [24] | 2021 | A survey on security and privacy of federated learning | | • | | • |
| [25] | 2021 | A review of FL for Healthcare Informatics | • | • | | • |
| [26] | 2020 | A review of applications in federated learning; industrial engineering to guide for the future landing application | | • | | • |
| [27] | 2021 | Federated Learning for Internet of Things: A Comprehensive Survey | • | • | | • |
| [28] | 2020 | Convergence of Edge Computing and Deep Learning: A Comprehensive Survey | • | | • | • |
| [29] | 2020 | Federated Learning for Vehicular Internet of Things: Recent Advances and Open Issues | • | • | | • |
| [30] | 2020 | Federated Learning in Mobile Edge Networks: A Comprehensive Survey | • | • | | • |
| [6] | 2021 | A Survey on Federated Learning Systems: Vision, Hype and Reality for Data Privacy and Protection | | • | | • |
| [13] | 2022 | Federated Learning for Smart Healthcare: A Survey | • | • | | • |
| [31] | 2022 | Applications of federated learning in smart cities: recent advances, taxonomy, and open challenges | • | • | | • |
| [32] | 2022 | Blockchain for federated learning toward secure distributed machine learning systems: a systemic survey | | • | | • |
| [33] | 2022 | Decentral and Incentivized Federated Learning Frameworks: A Systematic Literature Review | • | • | | • |
| [34] | 2022 | Federated Learning for IoUT: Concepts, Applications, Challenges and Opportunities | • | • | | • |
| [35] | 2022 | Federated Learning Approach to Protect Healthcare Data over Big Data Scenario | | • | | • |
| [36] | 2022 | Edge-Computing-Driven Internet of Things: A Survey | • | | • | • |
| [37] | 2021 | An Overview of Federated Deep Learning Privacy Attacks and Defensive Strategies | | • | | |
| [38] | 2022 | On the Edge of the Deployment: A Survey on Multi-Access Edge Computing | • | | • | |

### 3. Federated Learning

The FL system can be divided into several building blocks. The basic building blocks of the taxonomy include: communication architecture, federation scale, privacy mechanism, data partitioning, and machine learning models that can be used in FL. This distribution and summation are shown in the Figure 1.

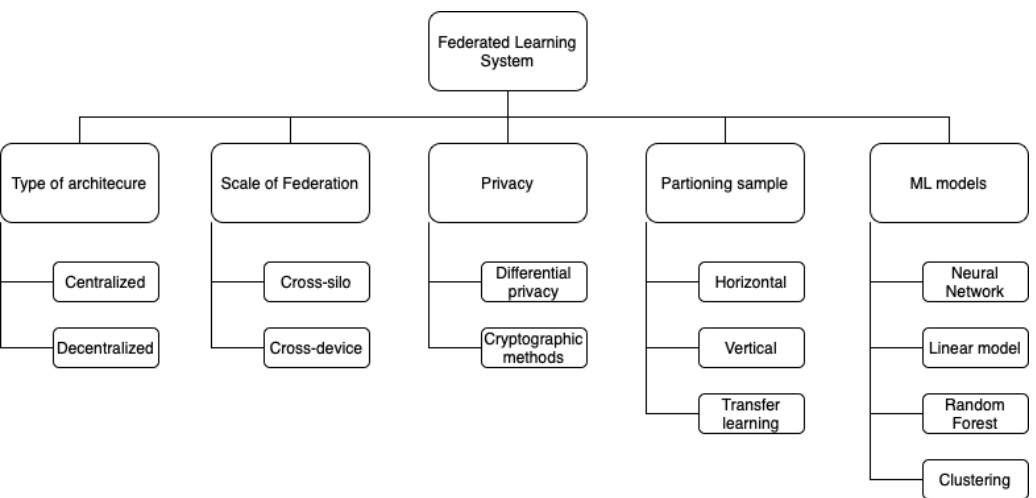

**Figure 1.** Taxonomy of FL system.

#### 3.1. Definition

The main characteristic of the federation is cooperation. Nowadays, it plays an essential part in computing and informatics systems, where multiple devices cooperate and communicate with each other. In computer science, federated computing systems are an attractive field of study in many state of affairs and contexts.

As mentioned in the introduction, FL is a set of devices that simultaneously participate in creating and training the model, using only their data for training, without the need to upload it and send it to the cloud and data centers. It covers techniques from various fields, including distributed systems, ML, and privacy. FL is a new ML strategy that tries to overcome the problem of data islands while preserving data privacy. It is well known that ML models perform better as the quantity and variety of trained datasets increase. Models based on such datasets may eventually lead to overfitting or limited scalability because the data supplied to a single organization (for example, a particular medical clinic) may be similar and poorly diverse. Constraints on privacy and secrecy significantly impact how well the ML model performs. FL is a repetitive process that runs in the required number of rounds. Since FL is an iterative process, models are not required to be close to convergence and fully trained to use it. FL consists of four basic iterative steps, shown in Figure 2. The first step is sending gradients from the clients to the central server. The second step is updating the global model on the central server. The third step is sending the global model to all clients who will participate in further training, and the last step is the local training, which is performed on the clients on their private data.

FL is an emerging machine learning system that aims to address the problem of data privacy. It involves multiple clients (such as mobile devices, institutions, organizations, etc.) coordinated with one central or multiple servers that are decentralized. Google first proposed it in 2016 to predict users' text inputs across tens of thousands of Android devices while protecting data consistency [26]. Google used Gboard keyboard and FL to train a neural language model for next word prediction, outperforming a model created using standard server-based collection and training. Language models have been trained with FL and differential privacy for additional privacy advantages. We can continue to improve the user experience while protecting privacy by using FL [39]. Several states, countries, and companies have been increasingly concerned about data privacy in recent years. Given

the importance of data available to any ML model, inventive solutions must be found to overcome limitations and allow model training to occur without the data ever leaving the location where it was collected and stored.

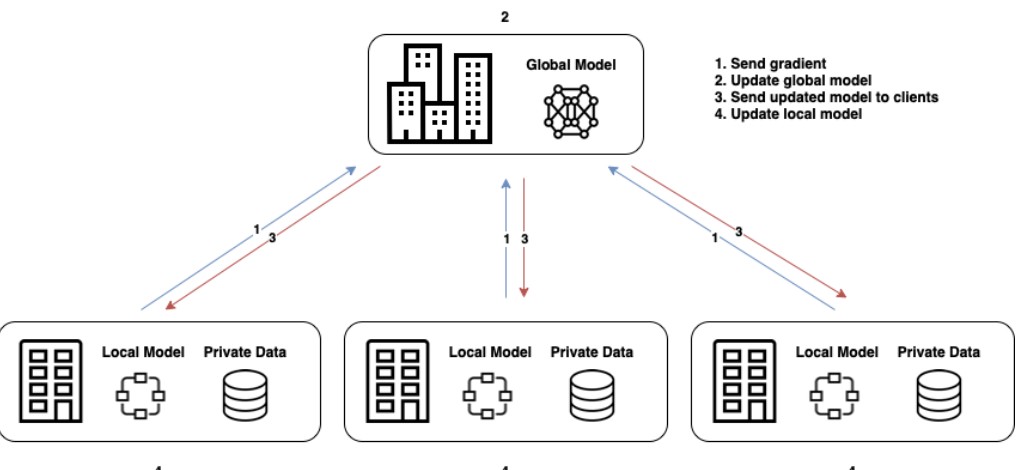

**Figure 2.** Architecture design of FL.

Today, much of the world's information is collected and owned on isolated islands rather than in central data centers. In other words, if they could be worked on where they are now, they could unlock much potential. In recent years, regulators in many countries have been paying more attention to the issue of data protection. Data security is critical nowadays. Given the importance of the data available for any ML model, it is necessary to devise inventive solutions to circumvent the limitations and allow the model to be trained without the data leaving the place of collection and storage.

### 3.2. Communication

The architecture design is shown in Figure 3, where three important parts of the FL are shown: FL server, FL clients, and communication-computation framework [40].

The FL server, sometimes called a manager, is usually a powerful computing device. The role of this server is to create a global machine learning model while managing the communication between clients and the server. In this case, the server should be reliable and stable to avoid deviations in creating the global model. If the server is unstable and unreliable, it may create an incorrect model. One possible solution may be to decentralize the federated server [41]. The use of blockchain can be used to solve these problems.

Blockchain technology [42] has a promising future. It can help business, government, logistics, and financial systems become more reliable, credible, and secure. Because it can help achieve these goals in various systems, this technology has many advantages. Of course, blockchain technology has some drawbacks, most of which are related to the cost and implementation process. Blockchain is constantly working to improve efficiency and reduce the cost of operation.

Nowadays, these two technologies are mentioned more often. For example, this paper [43] proposes the implementation of a coordination server for the FL algorithm to share information to improve prediction while ensuring data transparency and consents to its use. In parallel, the authors illustrate this approach using a prediction decision-support tool applied to a diabetes dataset. The authors present distributed K-means clustering based on differential privacy and homomorphic encryption, distributed random forest with differential privacy, and distributed AdaBoost with homomorphic encryption methods to enable multiple data protection in data sharing and model sharing. These techniques combine blockchain and FL to provide a comprehensive security assessment. The challenge may be finding a trusted server or party to serve as the administrator. Then, a fully decentralized environment where the parties communicate directly with each other and

contribute almost equally to the global training of the machine learning model seems to be the right choice.

In the case of FL clients, the clients are the owners of the data on which the ML model is built. All clients, also known as parties or data sources, have their own data and train their own local ML model. This local model is sent to the server, creating a global model. For clients, we need to consider their hardware capacity and storage. Client, i.e., an organization with significant computing power and servers involved in training a robust model, or also some small edge device whose performance may not be so sufficient. The same rules apply to mobile phones, which do not have much power and cannot train complex models.

In FL systems, computations are performed on clients, with clients and servers communicating. In most cases, communication and computation have the same goals: training the model and exchanging model parameters. While the clients are in charge of computing and creating the local model, the server is in charge of computing and creating the global model, which is then shared among the clients. Federated Averaging (FedAvg) [44] is a fundamental framework. The server first delivers the current global model to the chosen parties on each cycle. The chosen parties then update the global model using their local data. After that, the server receives the updated models once again. In order to obtain the new global model, the server averages every local model it has received. Until the set number of iterations has been reached, FedAvg repeats the process described above. The result is the server's global model.

### 3.3. Architectures

There are two significant types of architecture in FL: centralized and decentralized. FL [45], based on a centralized aggregation of local learning models for different IoT applications, is often considered. In this case, each device trains its local model on its private data. These updated local models are then sent to a central aggregation server. The central aggregation server is responsible for computing the global FL model. Figure 3 (Centralized) shows an example of FL architecture with a server to which clients are connected. This central server then aggregates this information for model updating purposes. Adversaries can still obtain some original data information, even if the data holders only make their abstract aggregated information available to others. The centralized server can be located at the network's edge or in the cloud so that clients can connect and communicate with it.

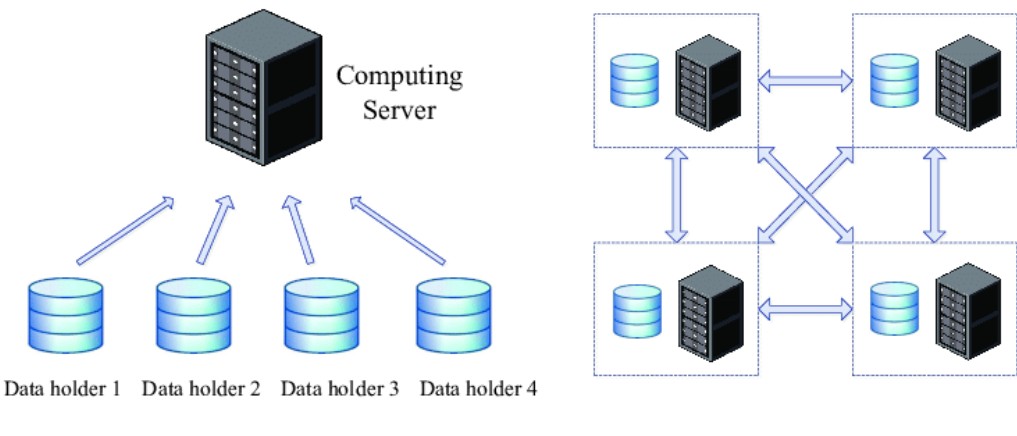

**Figure 3.** Architecture design of centralized FL vs. decentralized FL [46].

The article [47] describes the training and creation of a global model in IoT, which is based on a centralized server. The devices Raspberry Pi single-board computers wre used. The training aimed to be able to detect IoT intrusions. To evaluate the effectiveness of the proposed approach, the authors conducted rigorous experiments on the NSL-KDD dataset.

At the same time, the article compares FL training with a centralized server and centralized training, where the data is stored on a single computing machine. Centralized learning is so far defined as the most accurate. However, it has its disadvantages in terms of privacy and security. The results of several experiments demonstrate that FL with a centralized server is close in terms of performance and reliability to the results with the central approach of the methods used today as training models on a single machine. On the other hand, the choice of clients in FL significantly impacts the system's performance. Equipment failure during the process, long response time for uploading and updating models, and clients with less relevant data can limit and drastically reduce the accuracy of the global model.

Very similar research has been conducted in paper [48], where authors focused on several ML methods such as decision tree (DT), random forest (RF), support vector machine (SVM), and k-Nearest Neighbor (k-NN), as well as the top-rated deep neural network (DNN). This paper aimed to evaluate the accuracy of the models using FL with a central server and centralized learning for cyber-attack detection in IoT and IIoT sectors. All these measurements and experiments are then re-evaluated with similar results for all ML methods. Although a centralized design is commonly used in existing studies, a decentralized design is preferred in some aspects because concentrating information on a single server may pose a risk. While the centralized design is easier to implement, new studies are starting to emerge where the decentralized FL design is used.

However, the security of FL is increasingly being questioned due to constant attacks on the global model or user privacy data by malicious clients or central servers. New ways to prevent data leakage to third parties are constantly being explored to protect the data. Researchers have proposed several data privacy protection schemes within distributed ML architectures to avoid privacy leakage. Secure multi-party computation is a common approach. Each party or client uses ML techniques to build a local model on its private data. Because of this, FL is continuously improved and updated with new technological advances [46]. Three decentralized projects are available: peer-to-peer (P2P), graph, and blockchain. In P2P designed FL, the parties are equally privileged and treated equally [49]. The paper [50] discusses the design of a novel P2P algorithm for training machine learning models to detect non-IID anomalies, including mechanisms to locally rebalance training datasets by synthetically generating data points from a minority class. Other articles used P2P learning to train a collaborative model without needing a central node for better security and privacy [51,52]. For example, each party trains a tree sequentially and sends it to all other parties. The communication architecture can also represent a graph with latency and computation time constraints. Communication problems can arise in decentralized FL, so efficiency and communication among nodes must also be addressed [53]. Ghimire et al. [54] compare centralized and distributed learning in FL systems, concentrating on IoT cybersecurity within this context. Their analysis primarily focuses on security but highlights new difficulties and emerging research patterns. FL framework is based on a decentralized server, or the so-called blockchain [55]. Blockchain is new and quite popular and is developing very fast. Blockchain technology enables decentralization through the participation of servers within a distributed network. An approach is proposed to address these security issues [56]. Figure 3 (Decentralized) shows an architecture design where multiple servers are used to train and create the global model.

### 3.4. Scale of Federation

FL can be divided into two categories according to the scale of the federations: cross-silo and cross-device. The differences are based on the number of parties involved and the data each party stores. A relatively small number of trustworthy clients, typically businesses such as medical or financial institutions, corresponding to the cross-silo setting [57]. The larger organizations, companies, or data centers usually train the global model together. In most cases, these users are less in number but with a more significant amount of data stored in their data centers, [58]. One of the challenges facing such FL is

effectively distributing computation to data centers while adhering to the models' privacy constraints. Cross-silo does not have to be used only in medicine and financial institutions. A practical example can be the use of federated cross-silo learning also in the agro-food industry, where the authors focused on the optimization of soybean production and, at the same time, other cases of using this method in different problems' situations in the food industry [59].

The second category of FL mentioned is cross-device. In this case, a large number of devices participated in the training of the global model. However, these devices do not have an extensive dataset on which to train a given model nor a sizeable computational power that would allow them to train complex models. Since the devices are underpowered and cannot be relied upon for connectivity, the system needs to be sufficiently stable and reliable even in the event of a device failure or connectivity failure. A perfect example of cross-device use is the IoT or IIoT environment, where many devices, such as microcomputers and mini PCs, are used. Since these are edge devices that serve to control and manage machines, the devices themselves do not have so much performance, but they can perform some simple tasks in the framework of FL [60]. Another example may be mobile devices, which are increasingly numerous nowadays. In this case, connectivity problems arise because the Internet signal does not have to cover the whole territory, and at the same time, mobile devices can be in constant motion. In this case, we take advantage of the fact that several mobile devices are currently connected to the server, which collaborates on the training of the model [61].

### 3.5. Privacy and Security

With the development of big data technologies and high-performance computing, machine learning has opened up new possibilities for data-intensive science in the multi-disciplinary field of agri-technologies. In traditional ML, a centralized server's computing resources and training data are essential to the models' effectiveness and accuracy. Briefly put, in traditional ML, user data is kept on a central server and used for testing and training procedures that lead to the development of sophisticated ML models. In general, centralization-based ML approaches face several difficulties, particularly in terms of user data security and privacy as well as computational performance and time [62].

The security and privacy of the personal data collected are considered using AI-based technologies, following several legislative regulations protecting citizens' data and privacy, such as the General Data Protection Regulation (GDPR) [63] in the European Union. United States has the California Consumer Privacy Act (CCPA) [64] and the Personal Data Protection Act (PDPA) [65] in Singapore. This issue has become critical. They demand that personal data be processed transparently with a clearly stated purpose and the data subject's consent.

The topics of security and protection of personal data are nowadays the subject of more and more research. The same is true for FL, where many studies focus on security and information leakage in different segments and parts of FL. However, studies have demonstrated that adversaries can still compromise industrial applications and the manufacturing sector, medical and healthcare data in wearable devices, logs of personal data, and the decision-making of industrial robots or autonomous vehicles by improperly using shared parameters [66]. Often, the clients are the victims of an attack when the attacker tries to obtain the client's private data. FL was designed not only to train different models in a data-efficient manner but also to ensure user privacy, as their input data remains on the device and only individual weights and parameter gradients are shared. Because clients send updates to their local training parameters to the FL server, they are the most vulnerable attack surface in the FL system. In FL, the mention of the security and whether the sharing of gradients can be broken. Using these shared gradients, it is then possible to reassemble the image back into its original form or at least a similar one. Even multi-frame federated averaging on realistic architectures does not guarantee the privacy of all user data. Model aggregation is one of the most common privacy mechanisms in FL, which trains a global

model by summarizing model parameters from all parties to avoid transferring the original data in the training process. In [67,68], potential attack routes are discussed, the ability to reconstruct the input to any fully connected layer analytically is examined, a general attack based on optimization using cosine similarity gradients is proposed, and its applicability to various architectures and scenarios is discussed. According to experimental findings, privacy is not a property of inherent collaborative learning algorithms such as FL, and secure applications that potentially leak private information need to be thoroughly examined on a case-by-case basis. This reconstruction attack aims to iteratively add the generated small noises to the attack sample so that the generated gradient from this reconstructed attack sample approximates the actual gradient computed on the local training data [69]. The attackers can reconstruct the private local training data by analyzing the shared local gradient update parameters or the weight update vector.

As more and more demands are placed on privacy and security, current methods and algorithms are constantly improving and updating. Different options are emerging to address the gradient mentioned above attacks. Encryption of gradients and parameters is dealt with quite a lot nowadays, and this topic is discussed in detail. The main goal in these cases is that all involved clients send only encrypted gradients using homophobic encryption to a central computation server. The security of data storage is the main focus of general encryption schemes. Users without a key cannot perform computational operations on the encrypted data without risking a failed decryption and cannot extract any information about the original data from the encryption results. However, because homomorphic encryption addresses data processing security, it can address the issue of computing generally encrypted data. The ability for users to compute and process encrypted data without revealing the original data is homomorphic encryption's most significant feature. The processed data is simultaneously decrypted by the user who has the key. In this case, when the communication is encrypted, the actual learning and training process takes more time and requires more computational effort [70]. For cross-silo FL, some strategies and solutions significantly reduce the communication and encryption overhead caused by homophobic encryption [58]. A batch of quantized gradients is encrypted into a long integer and encrypted altogether instead of encrypting each gradient with complete precision. New quantization and encryption schemes are developed, as well as a new method for pruning gradients to perform gradient-based aggregation on ciphertext encoded batches. Other options aim to increase the security of the FL system. A good illustration is a federated accumulation system [71] based on blockchain and drones. In this case, a cuckoo filter is used to verify requests, and then a nonce timestamp is used to authenticate them. In addition, differential privacy is used to increase the clients' privacy.

*3.6. Categorization*

FL can be categorized into three groups based on the various sample space and feature space distribution patterns [72], as shown in Figure 4: horizontal FL, vertical FL, and federated transfer learning.

Horizontal FL is based on different data samples that share the same function or feature space. Two different users can collect data from different sources and create data samples. However, in the end, each user as a client can have a very similar data sample, even though the data is collected from different sources, and thus create a similar or identical feature space. Then, each client trains its local model on its data sample [73]. When two datasets' user features significantly overlap, but their users do not, horizontal FL is the best option. In horizontal FL, the datasets are divided horizontally, and a subset of the dataset is chosen for training whose user features are similar but whose users differ slightly from each other. For instance, two companies offer the same service in different areas, and their user bases are mainly unique. Nevertheless, the record user characteristics are also very similar because of their activities. In this case, horizontal FL can train the model, increasing the total number of trained samples and the model's precision. All parties typically compute and upload local gradients in horizontal FL so that the central server can combine them into

a global model [7,74]. A simple example is the already mentioned Google and its Gboard keyboard, where each Android phone user can train their local model on their data and then send it to a central server. The user is implied that even though the data may reside on the end user's device, it is consistent in feature space. Google GBoard is thus a cross-device, horizontal FL system used daily in real life.

Due to the different feature sets, vertical FL is also known as heterogeneous FL. When user features from the two datasets do not significantly overlap, but user overlap does, vertical FL is an option. By vertically dividing the datasets, FL removes the data portions where users are identical, but user features differ slightly from training. In other words, the same user is present in data across all columns. The feature dimension of training data can therefore be increased by vertical FL [72]. Vertical FL is more challenging to implement than horizontal FL because each machine learning algorithm requires a specific learning protocol to be created. Vertical FL is better suited for joint model training between large businesses because it has a broader range of applications [75]. For example, two businesses, one an e-commerce business and the other a bank, are located in the same city. They differ in user attributes while sharing a significant portion of user sets. In this instance, vertical FL is the most appropriate method for these two businesses to jointly train a distributed model, such as the risk management model.

Vertical and horizontal data partitioning can be considered components of federated transfer learning. When two clients have sparsely overlapping data samples and feature spaces, and we need to learn every sample label for a client, horizontal and vertical FL would be ineffective. In situations such as these, where the data sets from the two clients have different feature spaces and data samples, federated transfer learning is applicable. In other words, federated transfer learning is applicable when local client data may differ in data samples and feature space. Some frameworks allow a data scientist to train neural networks on data elements vertically partitioned among multiple owners, with the raw data remaining on the owner's device [76].

### 3.7. Summary

FL is a promising method that could solve many problems with traditional ML and AI techniques. With FL, it is possible to train a global model composed of many local client models. Each client participating in the training creates a local model using their dataset. As mentioned earlier, there are centralized and decentralized architectures. Each of these design architectures has advantages and disadvantages, such as the time and effort required for implementation or the security and protection capabilities of the system. FL cross-silo can be used in a variety of large organizations and institutions with sufficient data and computing resources, as well as in cross-device systems where many devices are used for training but do not have computing power. In these applications, edge devices and mobile devices are usually used. In all of this, it is necessary to ensure the protection and leakage of data. In this situation, it is required to ensure that no private information about the client is shared and that this information, as well as the communication between the client and the server, cannot be misused using various encryption and security methods.

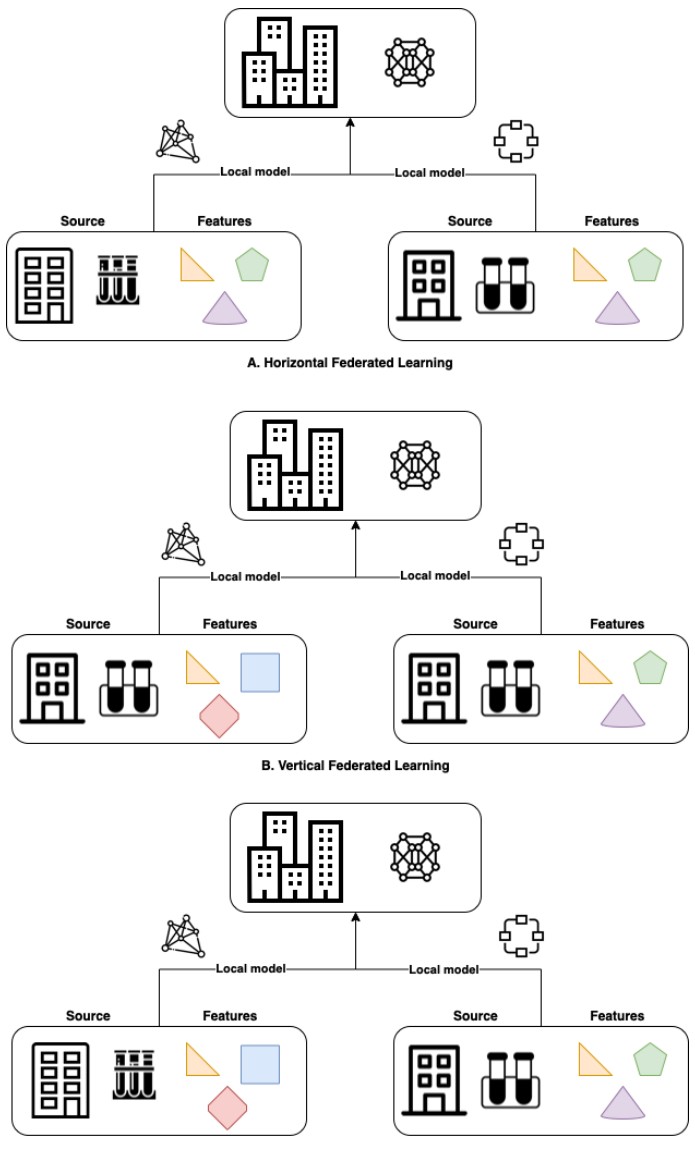

**Figure 4.** Data partition of horizontal federated learning, vertical federated learning, and federated transfer learning.

## 4. Frameworks

There are several frameworks and open-source software options available for FL. The right choice of framework depends on the purpose and type of use.

### 4.1. FATE

FATE (Federated AI Technology Enabler) [77] is an open-source project that aims to support a secure and federated AI ecosystem. It is an industrial-level framework developed by WeBank, a private-owned bank based in Shenzhen, China. It implements several secure computing protocols to enable significant data collaboration with data protection compliance. The overall structure of FATE can be divided into six categories: FederatedML, FATE Serving, FATEFlow, FATEBoard, Federated Network, and KubeFATE. FATE is one of the industrial exploitation projects that support institutions and larger companies in the creation of machine learning and artificial intelligence models. The advantage is the possibility of deploying this framework in a distributed way.

FederatedML is a library of federated machine learning. It contains and implements core machine learning algorithms and the necessary tools. For example, data preprocessing,

symptom engineering, and modeling are performed using federated machine learning algorithms. Tools also support federated machine learning, including several security protocols that securely compute multi-party interactions. A scalable, high-performance FATE Serving system for FL models is designed for production environments. The platform for the FL pipeline is called FATE-Flow. It is used for inference processing, modeling, training, verification, and publishing. FATEBoard is a tool to visualize individual ML models, explore them and understand them more easily. It provides different types of visualization and display of results in the form of tables or graphs. The FATE network is used for device-to-device communication and provides tools for developers and scientists to build algorithms using Federation APIs. The last element that contains the FATE framework is KubeFATE. This element is used to deploy FATE using cloud technologies such as Docker or Kubernetes. FATE provides excellent documentation and is easy to deploy in a real environment. The documentation states that FATE can be installed on Linux or macOS systems. Nowhere is it mentioned whether it also supports edge devices that work on ARM architecture.

We found a couple of articles that experimented with the FATE framework. The paper [78] studies the vertical structure of FL for logistic regression, where the datasets at two parties have the same sample identifiers but possess separate feature subsets. In the second case, [79], the authors' used machines with powerful hardware that contained several dozen CPU cores and a few hundred gigabytes of RAM. Logistic regression and XGBoost models were also used in this case. The authors focused on the cost per iteration and the time required per iteration within the FL and training machine learning model.

*4.2. Flower*

Despite algorithmic advances in FL, support for training FL algorithms on a device at the network's edge is still insufficient. This problem can be partially solved by the Flower framework [80]. It is a framework that is used for FL methods. Flower provides a novel method for conducting large-scale FL experiments and considers the highly varied FL facility scenarios. Flower was created as part of a research project at Oxford University. Many components can be extended or rewritten to create new modern systems that can be used later. As different machine learning frameworks have different strengths, Flower uses machine learning frameworks. Some famous machine learning frameworks include PyTorch, TensorFlow, MXNet, scikit-learn, TFLite, and raw NumPy. Flower allows for a wide range of configurations and scalability depending on the needs of each use case. Flower has a large number of users and a broad community. Therefore, this framework has excellent documentation and tutorials that allow rapid and easy deployment of this framework [81]. Flower provides several advantages, such as the scalability of a large number of parallel clients (support for up to 15 million clients), support and interoperability with different operating systems, hardware, and programming languages, and the advantage of deployment on mobile and edge devices with ARM architecture. This framework is also suitable for devices such as Raspberry Pi or NVIDIA Jetson Family, which are often used as edge devices. The user can freely extend, modify, and use Flower's extensive collection of built-in strategies, which include cutting-edge FL algorithms, for their experiments.

A couple of experiments focus on the Flower framework and test its capabilities and use in practice. An example is testing this framework on mobile devices with different Android operating systems and NVIDIA Jetson TX2 microcomputers. In the experiments, different local training rounds were compared, and different numbers of clients and their purpose and contribution to the global model result [82]. Kwing Hei Li et al. [83] evaluate the changes and communication load depending on the number of clients and the size of the model vector when using the Flower framework. They measured server and client CPU run times and total data transfer per client as the number of sampled clients increased.

### 4.3. OpenMined-PySyft

If there is a need to perform data science on data hosted on a PyGrid server, PySyft is a NumPy-based library that can be used for such tasks. PySyft is an open-source Python project for secure and private deep learning licensed from MIT. It is part of the Open-Mined ecosystem [84]. PySyft easily separates private data from training models using FL principles secured by various privacy-enhancing mechanisms. PySyft can only operate in simulation mode on its own. In order to support federated mode, it needs to be integrated with PyGrid and other OpenMined ecosystem projects. However, these initiatives are currently being actively developed, such as PySyft. PySyft intends to popularize privacy-preserving machine learning techniques by offering Python bindings and an interface reminiscent of conventional machine learning techniques. In order to support future break-throughs in privacy-preserving machine learning, PySyft seeks to be extendable such that new approaches such as FL, Multi-Party Computation, or Differential Privacy can be freely and quickly built and incorporated [85]. The goal of PySyft is to provide the best security for the clients that participate in the training of the global model. PySyft creates and uses automatic differential privacy to provide strong privacy guarantees to data users, independent of the machine learning architecture and the data itself. In this case, PySyft focuses on encryption and increasing privacy security for clients using homophobic encryption or encrypted computation like Multi-Party Computation (MPC). PySyft supports two types of machine learning libraries, namely TensorFlow and Pytorch.

In the OpenMined documentation, it was possible to find information about edge devices, specifically Raspberry Pi 4, and detailed installation and configuration procedures for FL. It also supported operating systems such as Windows, Linux, and macOS. FL is implemented via PyGrid on the web, mobile devices, edge devices, and other terminal kinds. PyGrid is an API for scaling up the management and deployment of PySyft. By using the PyGrid Admin application, it may be managed. Anirban Das et al. [86] focused on training CNNs on the RPI 4 device on MNIST data. They experimentally demonstrated FL using the PySyft framework on parametric study samples to compare the model convergence. Weight updates from clients are shared with the cloud to train the global model via FL. Similarly, Jiawen Kang et al. [87] used the PySyft framework in their study. In this case, however, they did not use edge devices but regular classic PCs with the Linux operating system.

### 4.4. OpenFL

Open Federated Learning (OpenFL) is a software platform for FL that was initially created as a component of a collaborative research project between Intel Labs and the University of Pennsylvania focused on FL for healthcare and medicine. Intel and the community that provides this framework are still developing OpenFL for general real-world applications [88]. OpenFL uses certificates to secure communication and provides scripts for deployment in bash. All participants must provide a valid public key certificate signed by a trusted Certificate Authority (CA) to establish communication and connection. The library consists of a collaborator or FL client that uses a local dataset to train global models and an aggregator or server that collects model updates and combines them to create a global model. OpenFL includes both a command-line interface and a Python API. The documentation describes that OpenFL is only available for Ubuntu operating systems version 16.04 or 18.04 and artificial neural networks built using TensorFlow or PyTorch might serve as an illustration. Additional ML model libraries and model training frameworks may also be supported through an extendable interface.

The first use of the OpenFL framework was directly in medicine in collaboration with the most significant international medical institutions, the aim of which was to gain insights and create a model for the detection of tumors in large and diverse patient populations [89]. Do Le Quoc et al. [90] address these challenges and propose SecFL. It is a confidential FL framework that uses Trusted Execution Environments. SecFL performs global and local training inside enclaves of Trusted Execution Environments to ensure the confidentiality and integrity of computations against strong adversaries with privileged access.

### 4.5. TensorFlow Federated

TensorFlow Federated (TFF) is an open-source machine learning framework for computations performed on decentralized data that provides the building blocks for FL based on TensorFlow [91]. This framework was developed for Python 3 for FL by Google. Google's main goal and motivation for creating this framework was the aforementioned mobile keyboard for Android phones, which predicted text for the user. It was the first example of using this framework in the real world. TFF enables researchers and developers to test new algorithms and simulate the built-in FL algorithms on their models and data. TFF can also be used to execute computations such as federated analytics that are not directly linked to model learning and training. The interfaces of TFF are split into two primary layers: Federated Learning API and Federated Core API. The FL API is an interface that makes it easier to carry out operations related to FL, model training, and model evaluation using TensorFlow. The interfaces offered by this layer consist of the following three parts: models, federated computation builders, and datasets. Federated Core (FC) can be understood as a subset of the lower-level interfaces that serve as the basis for the FL layer's implementation. In short, developers can implement its features in the FC. FC represents a development environment that allows compactly expressed program logic that combines TensorFlow code with distributed communication operators. These are the building blocks used in FL.

It was also possible to find articles and experiments comparing different frameworks for FL. In this case, [92] compares TFF and PySyft, which were used for text recognition. The findings demonstrate that federated text recognition models can match or even outperform central deep learning-based algorithms in accuracy. TFF achieves the best feature accuracy on a distributed dataset with five clients at 49.20%.

### 4.6. Other

Other less-known frameworks and libraries are also designed for FL. XayNet [93] is Xayn's FL backend. This open-source framework is designed for mobile devices and web browsers to train models. Hive [94] is the name of a FL system that Ping An Technology plans to create, but we have not been able to find official documentation. The Substra project is a closed-source solution that is currently under development. Although strategic discussions have been about the potential for opening up newer versions, no decision has yet been made [95].

Not all FL technologies are used in open-source frameworks. Leading companies have built proprietary libraries supplied under a restricted license and not open source. The NVIDIA Clara Train software development kit (SDK), which contains FL starting with version 2.0, illustrates such a library. NVIDIA Clara [96] is another framework that supports FL. It is a scalable computing platform. This platform was primarily designed for developers who create and deploy applications for medical, healthcare, and life science applications. The goal is to create intelligent devices and automate workflows. NVIDIA Clara provides libraries, SDKs, and reference applications for programmers, data scientists, and researchers. This framework's key contribution is GPU support and acceleration, which makes it possible to develop safe and expandable real-time applications. NVIDIA Clara supports several healthcare projects, such as Clara for Medical Devices, Clara for Drug Discovery, Clara for Smart Hospital, Clara for Medical Images, and Clara for Genomics. All these projects use artificial intelligence and NVIDIA Clara FL to improve the healthcare field. Several experiments use the framework mentioned earlier for FL. Most of these studies are focused on healthcare [97,98], but there is also research and experiments that deal with IoT and industry [99]. NVIDIA Clara used machine learning libraries, including TensorFlow and Pytorch.

IBM federated learning [100] focuses on data integration, customer privacy solutions, regulatory compliance, and big data across multiple locations. IBM FL is broadly configurable for various deployment scenarios, from mobile and edge scenarios and edge devices to multi-cloud environments in the enterprise to use cases that extend beyond the organization's boundaries. Data centers and cloud instances from various providers

frequently participate in FL, as do edge services with access to data from industrial facilities. IBM FL used a variety of machine learning libraries, including Keras, TensorFlow, Pytorch, SK Learn, and RLLib, which are compatible with IBM FL architecture [101]. Additionally, it offers APIs that can be used to create new algorithms for FL. IBM Federated Learning is independent of any particular machine learning library or paradigm. It can be used for deep neural networks or simpler algorithms such as decision trees or support vector machines. IBM provides two different versions of IBM: Federated Learning in IBM Cloud Pak for Data and IBM Federated Learning - Community Edition. Additionally, the IBM research team encourages fairness strategies for FL that lessen bias [102].

*4.7. Summary*

Developers are constantly improving their frameworks to provide users with as many benefits as possible. One of the most comprehensive is FATE, which supports many ML methods and algorithms in both horizontal and vertical settings and a large number of tools that make the programmers' work more accessible. Flower and TFF are open-source frameworks that are very simple to implement. The OpenFL framework from Intel, in cooperation with the University of Pennsylvania, provides users with high security against threats and data leaks, where they require different kinds of certificates for communication. Similarly, PySyft strives to provide users with maximum protection for their software. IBM Federated Learning is suitable when communicating with cloud platforms and in cases where there is a large amount of data to be processed. However, it also supports mobile and edge devices. The last mentioned NVIDIA Clara, suitable for IoT applications, is mainly used in medical and healthcare.

Each of these frameworks has advantages and disadvantages. They may differ in the supported devices, operating systems, data security mechanisms, or communication speed among devices. Information about these frameworks is also summarized in Table 2. Some experiments have been carried out with some frameworks, where the accuracy of individual frameworks and their speed in training different ML methods were compared [40]. As was already mentioned during the literature survey, in this case too there were problems in supporting frameworks and ARM architecture in many cases, because the documentation did not always indicate whether the given device is supported or not. We also did not read this information in other articles and we did not find any relevant information about it. We also looked at the possibilities of deploying these frameworks at the network's edge and on edge devices. Since many edge devices can use the ARM architecture, it is important to note that not every framework supports this type of architecture. Of the above frameworks, only Flower and PySyft have official support for ARM architectures, according to the documentation.

**Table 2.** Supported features of FL frameworks.

| Framework Supported Features | | FATE 1.5.0 | Flower 1.0.0 | PySyft 0.6.0 | OpenFL 1.3 | TFF 0.29.0 |
|---|---|:---:|:---:|:---:|:---:|:---:|
| OS | Linux | • | • | • | • | • |
| | MacOS | • | • | • | • | • |
| | Windows | | • | • | • | |
| | iOS | | • | • | | |
| | Android | | • | • | | |
| Models | Neural network | • | • | • | • | • |
| | Linear models | • | • | | | |
| | Decision tree | • | • | • | | • |
| Data partitioning | Horizontal | • | • | • | • | • |
| | Vertical | • | | • | • | |
| Settings | Cross-silo | • | | • | • | • |
| | Cross-device | • | • | • | | |
| Support ARM architecture of edge devices | | | • | • | | • |

## 5. Applications

FL can be applied in several real-world situations. This section provides examples of situations where FL may be employed in the future. FL is particularly useful in contexts that prioritize security and data sensitivity. Early users identified FL's great potential and started several related investigations and attempts to put FL to practical use, despite FL's limits and significant difficulties, particularly in the field of security.

### 5.1. Mobile Devices

Google, as mentioned above, designed and came up with the concept of predicting and forecasting user input using the GBoard keyboard. This keyboard is constantly being improved and updated. It supports more than 900 language varieties [103]. In the same way, emoji prediction is currently used. Based on the text input from the user, the keyboard evaluates and offers the user which emoji is suitable. This keyboard is already deployed and used daily [104]. Apple is also advancing ML and DL and using this expertise in speech recognition. As this data is also classified as sensitive and proprietary, FL is used in this case for the training and classification of the model. The model uses the features extracted from the phrases that trigger the speaker verification system [105]. Apple also focuses on the development and training of several languages that are most commonly used around the world [106].

Although mobile device storage and processing power are expanding rapidly, it is challenging to meet mobile consumers' rising demand for quality due to communication bandwidth limitations. Most comprehensive providers offer service environments at the mobile network's edge, near the consumer, instead of integrating cloud computing and cloud storage to reduce network congestion. Mobile edge computing (MEC), nevertheless, carries a more significant risk of data loss. Of course, this field is also constantly working on improving the quality of service and ensuring privacy from data leaks. Several studies focus on mobile devices using FL. Feng et al. [107] propose a privacy-preserving mobility prediction framework via FL, using a central server where no private and sensitive data is uploaded. Wang et al. [108] also give another example of FL usage in mobile devices. The goal is to intelligently leverage collaboration among devices and edge nodes to exchange learning parameters for better training and inference of models. They propose to make edge nodes and mobile devices more intelligent, and optimize computation, caching, and communication. Qian et al. [109] propose a privacy-aware service placement scheme to address the problem of service placement with privacy-awareness in the edge cloud system using FL.

### 5.2. IoT Systems

As with mobile devices, the use of FL is becoming more and more common in IoT systems and solutions. Due to the exponential expansion of IoT devices and the considerable data dispersion of large-scale IoT networks, AI functions are frequently located in data centres or cloud servers in traditional IoT systems, which are not scalable. Due to the growing number of edge devices, the higher performances can handle more complex processing tasks and thus relieve the computation on the clouds. The smart home is one example and area where IoT can be applied. In order to better learn user preferences, devices in smart home architecture would upload some related data to the cloud server, which may lead to a data breach. Nowadays, many IoT projects in industry, smart homes, smart cities, and robotics use a lot of data from smart sensors. In this case, FL can be used to train a model to detect anomalies. Nguyen et al. present an autonomous self-learning distributed system for anomaly detection of IoT devices [110]. The work [111] focuses on anticipating user behaviour in smart homes and provides a straightforward ML model with a temporal structure to achieve a decent trade-off between accuracy, communication, and computing cost.

### 5.3. Industry

There are many cases where FL can be used in the Industry 4.0 or Industry 5.0. It makes sense for industrial engineering to adopt FL applications in response to FL's success in protecting data privacy. Visual object detection is an artificial intelligence technique based on computer vision that has many practical applications, e.g., fire hazard monitoring. Liu et al. [112] proposed a platform that was deployed in a collaboration between WeBank and Extreme Vision to help customers facilitate computer vision-based safety monitoring solutions in smart city applications. Han et al. [113] and Xia et al. [114] addressed industrial IoT, considering that most of today's industrial smart factories and machines utilize ML-based models. They analyzed and summarized how to process a large amount of data and their subsequent use in automotive, energy, robotics, agriculture, and healthcare. Ramu et al. [13] explore the possibilities of FL and digital twin in different deployment scenarios, e.g., smart cities. Hua et al. [115] used blockchain-based FL to implement asynchronous collaborative machine learning between distributed clients that own the data. The authors focused on real historical data available from current railway systems. Deng et al. [116] are concerned with using FL to manufacture components and structural parts for aerospace companies. In this case, they address the problem of labor-intensive, limited data from a single enterprise and the high cost of data mining. They are looking for clients with similar data sources to solve this problem. Subsequently, a neural network is proposed to be used in the model training process.

### 5.4. Healthcare

As a disruptive method of data privacy, FL holds great promise in healthcare. Every healthcare institute may have a wealth of patient data, but it may not be enough to train its prediction models. At the same time, usually, no medical institute can further share this data for further processing due to privacy and data protection concerns due to health regulations. The combination of FL and disease prediction is one of the promising solutions to break the barriers of analysis in different hospitals and institutions.

Many studies have also been carried out within the medical domain on protecting patients' health or predicting diseases. Pfohl et al. [117] investigated differential private learning for electronic health records via FL. Their results found that the performance is comparable to centralized learning. Kumar and Singla [118] write on the demonstration and adaptation of FL techniques for healthcare. They introduced and described areas of electronic systems such as drug discovery and disease prediction systems. A community-based FL algorithm was put forth by Huang et al. [119] to forecast the mortality and length of hospital stay. Electronic health records are grouped into communities within each hospital based on standard medical characteristics. Instead of a universal model shared by all hospitals and consequently all patients, each cluster learns and uses a unique ML model customized for each community, improving its performance and efficiency. Using electronic health records' data dispersed across various data sources, FL is also used in [120] to forecast hospitalizations for patients with heart disease during a target year. In [121], a distributed system is designed and implemented to address intrusion detection in medical cyber-physical systems. Processes of private data from patients' devices (e.g., heart rate, blood oxygen saturation, etc.) are locally trained to improve the global model. which is then used to detect harmful activities. Homogeneous patients with similar characteristics are grouped to provide a high-throughput model. Each cluster creates its personalized local and global model.

### 5.5. Finance

The financial sector is also one of the sectors where the highest possible security of private client data is required. Thanks to banking, individual customers can now own their banking information, which is crucial for developing a new ecosystem of data markets and financial services. For banking, FL needs to be modified and improved to address several

real-world issues, including managing user incentives, limiting user access to personal data, limiting user scope, and limiting user time.

Long et al. [122] introduce the use of FL in open banking, where they focus on and discuss statistical heterogeneity, model heterogeneity and access limits within the banking system. Shingi [123] describes the problems in the bank loan approval process. Many times, this process has to be approved manually, and banking institutions do not have enough data to automate these processes and create machine learning models to make these decisions. The lack of data causes inaccurate models that are unusable in the real world. In this case, FL can solve the problem of data scarcity, as more financial institutions could be involved in training a global model. In [124], they analysed and presented the credit risk assessment process. The aim is to propose a model using FL to predict the risk arising in the case of credits. Currently, the Chinese bank WeBank [125] is trying to create various systems and processes that would simplify the bank's decision-making. The bank uses various types of encryption in network communication to prevent and reduce the risk of data leakage. They have implemented the FedRiskCtrl mechanism [126]. This FL mechanism evaluates loan applications for small and medium-sized enterprises.

### 5.6. Transport

Vehicle networks have become more robust due to recent developments in sensing and communication technologies and the amount of data coming from in-vehicle sensors, embedded devices, and road cameras. Increasingly, AI and ML methods are also being used in transport to create intelligent transport systems. By directly running ML models in vehicles based on their datasets, such as road geometry, collision avoidance, traffic flow, and data from all the sensors available in modern vehicles, FL can support intelligent traffic systems. Since modern vehicles have many sensors at their disposal, they can generate a large amount of data that could be used to train a global model. FL would simultaneously address the protection and security of this data, and at the same time, this approach could be integrated with other computing power at the edge and IoT.

The joint optimization problem of choosing a communication interface and the problem of choosing a route based on cooperative learning between vehicles can be solved using FL [29]. FL could facilitate the development of different types of IoT applications in vehicles. For example, in autonomous driving systems, each vehicle is trained online based on observations of a single vehicle, leading to limited knowledge of the environment. FL can provide more information for each vehicle through inter-vehicle collaboration. Chen et al. [127] focus on using FL in autonomous vehicles. They propose a decentralized method for autonomous vehicles and simultaneously demonstrate through experiments that their FL-based methods are suitable for improving multi-object recognition performance in autonomous vehicles. Zeng et al. [128] focus on a novel FL framework that enables large-scale wireless connectivity for designing controllers of autonomous vehicles. The autonomous vehicle system must be continuously improved to react to unpredictable things in real-time and promptly, to react automatically to acceleration and deceleration, braking and stopping, or getting into the right lanes. Elbir et al. [129] explore applications within vehicle networks and the development of intelligent transport systems such as autonomous driving, infotainment, and route planning.

### 5.7. Summary

Table 3 summarizes the experiments and applications that can be implemented using FL. As can be observed, FL can be used in a wide range of applications. Just as it has found applications in mobile devices, the use of FL for security is increasingly required in the financial and healthcare sectors, where it is impossible to share sensitive patient and client data to clouds and data centers. In these articles, it is not always stated and clearly defined on which devices a given model was trained and which devices it consisted of. In the case of mobile devices, these are mostly Android and iOS phones. For other applications, authors

did not always specify what types of FL devices were trained and which devices were communicating. In many cases, there was not any hardware nor software specifications.

**Table 3.** Overview of federated learning applications.

| Source | Year | Application Type | Description |
| --- | --- | --- | --- |
| [103] | 2019 | Mobile devices | Google GBoard for Andriod phones for predicting and forecasting user input |
| [104] | 2019 | Mobile devices | Keyboard to predict emoji based on text entered by the user |
| [105] | 2020 | Mobile devices | Apple uses FL to train a model for speech recognition |
| [107] | 2020 | Mobile devices | A privacy-preserving mobility prediction framework based on phone data via FL |
| [108] | 2019 | Mobile devices | Intelligent use of cooperation among mobile devices and edge nodes to exchange parameters for optimization of calculations, caching, and communication |
| [110] | 2019 | IoT systems | Autonomous self-learning distributed anomaly detection system for IoT devices |
| [111] | 2020 | IoT systems | It focuses on predicting user behavior in smart homes and provides a simple ML model with a time structure to achieve a decent trade-off between accuracy, communication, and computational cost |
| [130] | 2021 | IoT systems | Survey and oveview provided new insights into applications in IoT, development tools, communication efficiency, security and privacy, migration, and scheduling in edge FL |
| [45] | 2021 | IoT systems | A set of metrics such as sparsity, robustness, quantification, scalability, security, and privacy are defined to rigorously evaluate FL, proposing a taxonomy for FL in IoT networks |
| [131] | 2022 | IoT systems | Anomaly detection on edge devices that can detect disruptions in IoT networks |
| [132] | 2019 | IoT systems | Design and use of the FL framework for joint allocation of communication and computational resources |
| [112] | 2020 | Industry | The platform, which was deployed in a collaboration between WeBank and Extreme Vision to help customers develop computer vision-based security monitoring solutions in smart city applications |
| [113] | 2019 | Industry | Utilize FL in an industrial setting for tasks related to visual inspection of products; this approach could address manufacturing processes' weaknesses while offering manufacturers privacy guarantees |
| [114] | 2018 | Industry | Analysis and processing of large amounts of data and its subsequent use in the automotive, energy, robotics, agriculture, and healthcare industries |
| [13] | 2022 | Industry | Deployment possibilities of FL and its digital twin in different deployment scenarios, such as smart cities |
| [115] | 2020 | Industry | FL based on blockchain to implement asynchronous collaborative machine learning among distributed clients that own the data, with real historical data available from network systems |
| [116] | 2022 | Industry | Use of FL in the manufacture of components for aerospace companies |
| [117] | 2019 | Healthcare | Differential private learning for electronic health records via FL |
| [118] | 2021 | Healthcare | Demonstration and adaptation of FL techniques for healthcare and electronic systems in healthcare such as drug discovery and disease prediction systems |
| [119] | 2019 | Healthcare | FL community-based algorithm to predict mortality and length of hospital stay; electronic health records are clustered into communities within each hospital based on standard medical characteristics |
| [120] | 2018 | Healthcare | Forecast hospitalizations for patients with heart disease during a target year |
| [121] | 2019 | Healthcare | Implemented a distributed system to address intrusion detection in medical cyber-physical systems |

**Table 3.** *Cont.*

| Source | Year | Application Type | Description |
|---|---|---|---|
| [122] | 2020 | Finance | The use of FL in open banking, where statistical heterogeneity, model heterogeneity, and access constraints within the banking system are addressed |
| [123] | 2020 | Finance | Describes the problems in the process of approval of bank loans and at the same time the problems of a lack of data; the use of FL and the involvement of several financial institutions involved in the training |
| [124] | 2019 | Finance | Analyze the loan risk assessment process; propose a predictive model using FL to predict the risk arising in loans |
| [125] | 2022 | Finance | Create various systems and processes to simplify the bank's decision making |
| [127] | 2021 | Transport | Development of autonomous vehicles, how they communicate with each other and how they drive; acquisition of data from multiple vehicles simultaneously and subsequent training of a global model |
| [128] | 2022 | Transport | Autonomous vehicle system; reacting to unpredictable things, automatically responding to acceleration and deceleration, braking and stopping, or getting into the right lane |
| [129] | 2020 | Transport | Applications within vehicle networks and the development of intelligent transport systems such as autonomous driving, infotainment, and route planning |

## 6. Challenges

In this case, it is necessary to focus on the challenges that need to be addressed in the EC and FL. Several constraints and problems can arise with devices at the network's edge. This section will describe the different basic directions of FL research and the outstanding issues in AI, ML development and deployment, security, and data protection at the network's edge.

### 6.1. Edge Computing

Nowadays, there are a large number of devices that provide different computational power. We can divide these into two parts: cloud or edge paradigm, which are based on the computational performance and location of individual computing devices. Both paradigms are nowadays used for different application scenarios. The distribution of computing resources and processes, while bringing them closer to data sources through edge computing, is a more popular paradigm for computing that helps address privacy and security issues. The biggest advantage of edge computing is the ability to reduce costs while still maintaining maximum operational effectiveness [133]. The processing of data at the edge facility eliminates the need to move it to a central service provider and allot large amounts of storage space, resulting in cost savings with high efficiency. Another benefit, especially in situations where real-time response is required, is that computations are performed at or near the edge, on or near edge devices, and are faster than those performed through cloud computing [134]. Edge computing, which describe devices located close to the end devices, has a great deal of potential to deliver the low-latency, energy-efficient, secure services that are now essential in today's society. Due to limited computing and power resources at the network's edge, exploiting these facilities for AI and ML processes was impossible. Cloud computing is commonly used to train demanding ML models on powerful devices and graphics cards connected to a dedicated cluster. The Table 4 summarizes the basic comparisons of edge and cloud computing.

**Table 4.** Edge and Cloud computing.

| Factor | Edge Computing | Cloud Computing |
| --- | --- | --- |
| Computing resources | Low computing resources with limited performance, memory, and storage space | High computing performance, available memory and storage resources |
| Data | Stored locally on edge devices | Stored centrally in data centers |
| Latency | Real-time data processing | Data processing that takes a long time and is not time-urgent |
| Connectivity | Problems with internet connection, limited internet connection | The need for a reliable internet connection without interruptions |

### 6.2. Hardware Requirements

An edge device can be defined as a device with limited computational, memory, and power resources and cannot be easily scaled up or down. These limitations may be due to size or due to cost. This definition of edge device applies to all such consumer and industrial devices, where resource constraints limit what is available to build and train AI models [135]. Such edge devices with limited resources can be found in today's IoT and industry world, which are used, for example, for environmental monitoring, in smart cities, or smart factories. Edge devices are used in these IoT environments because they are more compact. Drones of all varieties, smartphones, and other edge devices with constrained resources that communicate remotely can be categorized as such. A summary of edge devices that can be used at the edge of the network can be found in [136]. In this summary, it is possible to observe the various hardware and software differences of these devices, which can cause significant problems in FL. At the same time, extensive research is being carried out in orchestrated end-edge-cloud architectures for flexible AI-of-Things systems [137]. Our goal is to focus on the possibilities of deploying AI, but also on creating new models at the edge of the network on devices such as Raspberry Pi, Jetson Family, various edge TPUs or mini PCs. In this case, these are devices with limited computing power, which often communicate wirelessly, their performance is not high, they have limited memory resources, the heterogeneity of the devices can be very wide, and there is a need to ensure the highest possible data protection and privacy. ML and AI are capable of performing a wide variety of advanced tasks such as image classification, object detection, or sound recognition. These tasks often require machines that have high computing power, large demands on memory, and storage space, and thus have a high consumption of electricity.

In this context, it is necessary to research the most recent EC hardware requirements and the gaps that need to be filled before implementing FL in the EC paradigm. This concept motivated us to conduct a simple survey of EC hardware specifications. The devices that can be used at the edge of the network are summarised briefly in the Table 5. In EC, it is possible to use several types of devices that can be deployed at the edge of the network:

- Single board computers (SBC): The SBC is an entire computer with a microprocessor, memory, input/output pins, and other features needed for a working computer that is built on a single circuit board. The ARM architecture is used by many manufacturers. The ARM architecture is also present in today's smartphones, tablets, and laptops. Compared to desktop or laptop computers, SBCs are quick to produce and reach the market. Compared to multi-board computers, they are lighter, smaller, more reliable, and more effective [138];
- Mini-PC: Compact computer or mini-PC systems are ideal for running and training simpler AI and ML algorithms, real-time analysis of high data flows, and simpler image processing from numerous sensors. The advantages include the processing speed, compact design, and cheaper price. Despite having a relatively high processing power, these computers frequently experience cooling issues because all of their components are contained in a small box with a subpar cooling system. In some circumstances, it is possible to swap out some parameters for newer, more powerful

ones, such as the CPU, RAM, or hard drive. However, the majority of these computers only have slower graphics cards or graphics cards integrated with CPU [139];

- Field Programmable Gate Array (FPGA): A type of reconfigurable integrated circuit known as a FPGA is composed of programmable logic blocks, each containing gates that can communicate with one another. This device's advantage is that it can be reprogrammed to alter how each block's logic operates and the connections between them. FPGAs are now frequently used for edge nodes. Although they are not as powerful as GPUs used for complex computations, they are more efficient and use less power. These devices create new possibilities for utilizing these accelerators at the network's edge [140].

**Table 5.** Overview of hardware requirements.

| Type | Manufacturer | Pros | Cons |
|------|-------------|------|------|
| SBC | Nvidia Jetson Family, Google Coral, Rasperry Pi, Bearkey, Bitmain | ARM architecture with very low power consumption, compact size, possibility to power devices from batteries | Very low performance compared to other listed devices, limited, computing and memory resources |
| Mini-PC | Intel or AMD | Solid performance for processing simple AI and ML models, processing speed, lower price | Problems with cooling system, lower efficiency, and higher power consumption |
| FPGA | Intel, Xilinx, Microchip Technology | High performance per watt of power consumption, lower costs for large-scale operations, and good performance for calculations and tasks that require more complex calculations | Requires a significant amount of storage space and external memory, programmed for only one specific task |

### 6.3. Communication Overhead

One of the main issues in the FL-based IoT environment is communication overhead. The large amount of data transferred during the process and the iterative, non-optimized method of communicating between the server and the clients are the main causes of the increased communication overhead. Reducing the number of communicating devices, the total number of communication rounds, and the size of the model update are the keys to reducing the communication overhead in FL in the edge network. However, the global model may have a lower accuracy if the number of clients are reduced. The process of reducing the size of model update messages sent between server and client is known as model compression. In the FL implementation, compression methods such as compression quantization and partitioning are used to minimize the update size. When clients lack resources, this issue becomes unfavourable. For instance, a client could not successfully connect with the FL server during model training if the client had limited bandwidth. The same case can occur if the client has insufficient data processing capability. Most of the existing research focuses on reducing the total number of bits transferred for each model update [141]. Wang et al. [142] focus on an orthogonal approach that identifies irrelevant updates by clients and avoids sending them to the server to reduce the network overhead. Wu et al. [143] present the FL method called FedKD. As part of this method, they strive for more efficient and effective communication. This method is based on the adaptive mutual distillation of knowledge and dynamic gray data compression. There are various review articles that are dedicated to communication and efficiency in FL. On the one hand, FL plays an important role in the resource optimization of wireless communication networks, on the other hand, wireless communication is crucial for FL. Thus, there is a communication between FL and wireless communication, which is complicated in some cases. An example can be [144], where the authors discuss in depth the bidirectional communication between the client and the server and discuss the possibilities of optimizing and making this communication efficient. Liu et al., in their [145] review, address in-vehicle task offloading using edge and vehicular clouds. They aim to identify several open

problems and challenges for future improvement of task-based reinforcement learning. Related to this are the possibilities and ways of communication in wireless networks with edge servers.

### 6.4. Limited Resources

Heterogeneous FL clients' limited power or memory capacity and energy consumption can cause specific issues. Each FL client might only have a tiny amount of memory, and a client with more memory might run more complex calculations. Additionally, FL clients' preconfigured energy budgets might not be sufficient to meet system demands throughout the training session. Lack of memory causes devices to overflow, while limited computing capabilities necessitate longer processing times. These factors resulted in increased communication overhead and reduced system performance, which can cause significant problems when training models. Nishio and Yonetani [146] focus on heterogeneous devices with different computational power. Their goal is to efficiently perform computations on these devices and thus reduce the computation time on lower-performance devices or reduce the upload time in the case of poorer connection quality. Hard et al. in [147] have focused on the necessary hardware requirements, required memory size and computational capability while implementing next-word prediction on the keyboard. Jiang et al. [148] are concerned with developing the PruneFL approach, which aims at adaptive and distributed parameter trimming and model size adjustment. During this process, the model size is adapted. The first experiments demonstrate that even on devices such as Raspberry Pi, it can be observed that the training time of the model is reduced. At the same time, the authors show that the accuracy of their model is very similar to the baseline model, without parameter trimming and model fitting.

### 6.5. Heterogeneous Hardware

A wide range of heterogeneity device can be involved in the FL process. In this case, it may not be only other device performances but also different operating systems, different device generation, or different data sizes on which the local model is trained [149]. Therefore, the training period may vary significantly from client to client, and it is not efficient to consider all participants with the same range [150]. To obtain optimal training results, the FL must be aware of heterogeneous hardware configurations and be able to adapt training on different devices based on this. From the connected clients, selecting only the most reliable ones for their system requirements is always necessary. To overcome these difficulties, Feng et al. [151] research the use of wireless power transfer and heterogeneous computing in FL. They suggest a heterogeneous computing and resource allocation framework based on heterogeneous mobile architecture to achieve effective FL implementation. He et al. [152], in their paper, present the possibilities of training adaptation for heterogeneous devices. Their formulation of the training efficiency maximization problem, in particular, establishes a novel analytical relationship between training loss, resource consumption, and heterogeneity. Their first experiments and simulations yielded training speedups of 5–15% at different heterogeneities.

### 6.6. Heterogeneous Data

With heterogeneous hardware, it is possible to talk about different types of devices that participate in the training, which can significantly affect the whole training process and the accuracy of the global model. However, there can also be a problem when it is possible to talk about heterogeneous data or Non-IID data [5]. The training data of each client in FL depends to a large extent on the use of specific local devices, and therefore the data distribution of connected clients may be quite different from each other. In this situation, it is equally important to solve communication and optimization problems that eliminate these differences in the data. Federated optimization has several key characteristics that distinguish it from a typical distributed optimization problem. Non-IID training data on a given client is typically based on using an edge device by a particular user. Therefore, a

specific user's local dataset will not represent the population distribution. Data imbalance can be another problem. Some users may have more possible data and a more significant local data sub-set than users who will have fewer data. This can significantly affect the speed of the whole training process. There are studies that focus on non-IID data. Zhao et al. [153] have focused on experiments that demonstrate the extent to which non-IID data can affect the accuracy of the global model. The accuracy of the FL and the global model can be significantly reduced if the local data is of non-IID type. In some cases, the accuracy can drop by 55% in the case of neural networks and highly skewed non-IID. One of the most famous algorithms used in training is the FedAvg algorithm. However, it is known that this algorithm is not very good at dealing with non-IID data and these data can significantly affect the output of the global model when using FedAvg [154]. There are studies that try to eliminate the problems that arise with FedAvg. Based on this, a couple of new algorithms have been developed based on the FedAvg algorithm. Wang et al. [155] provide a general estimation and analysis of the convergence of federated heterogeneous optimization algorithms. They provide a principled understanding of the distortion of the alignment and the slowdown in the convergence of the global model due to data consistency. As part of this research, they propose the FedNova algorithm, in which they focus on a normalized averaging method that eliminates object inconsistency while preserving fast error convergence. This algorithm solves heterogeneity problems in federated networks. The authors tried to generalize the re-parametrization that is present in FedAvg. Initial experiments of this algorithm show that it is significantly more stable and accurate under convergence management than FedAvg. Another known algorithm is also the SCAFFOLD [149]. Karimireddy et al. focus on reducing the variance from the data collection in their local updates. They demonstrate experimentally that SCAFFOLD requires significantly fewer communication rounds and is not affected by data heterogeneity or client sampling. They also bring faster convergence. There are many articles, studies and researches that focus on comparing individual algorithms, which are summarized in the Table 6.

**Table 6.** Algorithms overview.

| Source | Year | Type of Algorithms | Description |
|--------|------|--------------------|-------------|
| [156] | 2022 | FedAvg, FedProx, FedPD, SCAFFOLD, Fedmed | Analysis of heterogeneity in FL and overview of practical problems caused by systemic and statistical heterogeneity |
| [157] | 2022 | FedDM, FedAvg, FedProx, FedNova, SCAFFOLD | Experimental tests that compare FedDM with multiple algorithms that perform image classification, model performance, and model accuracy |
| [158] | 2022 | FedAvg, FedProx, FedNova, SCAFFOLD | Overview and design of comprehensive data distribution strategies to cover non-IID cases. Extensive experiments to evaluate state-of-the-art algorithms currently in use |
| [159] | 2021 | FedAvg, FedBCM, FedPer, FedDF, FeSEM | A detailed analysis of the influence of non-IID data on parametric and non-parametric machine learning models in both horizontal and vertical learning. |
| [160] | 2022 | FedAvg, FedCurv | Empirical behavior in common scenarios on non-IID data and experiments that can lead to increased performance and reduced communication costs |

### 6.7. Scheduling Techniques

Due to the wide use of computationally intensive tasks and applications, it is necessary to address the scheduling of tasks. Existing scheduling optimization techniques can be divided into synchronous and asynchronous. Each training round synchronous communication, a subset of clients, is triggered to train the local model. Problems with device performance or network reliability may result in some clients not responding and communicating with the server at the requested time. In this case, the server keeps waiting for a response until it receives a response from a sufficient number of clients. Otherwise, the server must discard this epoch and continue with the next one [161]. Sprague et al. [162] introduce asynchronous communication between the clients and server and study its con-

vergence speed when distributed on multiple edge devices. Asynchronous optimization, in this case, allows FL participants to send gradients directly to the FL server after each local update, which is not possible with synchronous FL optimization. Asynchronous communication is nowadays the subject of several studies, because this type of communication is one of the fastest and most reliable. After all, convergence is faster in this case, and at the same time, this type of communication is less sensitive to heterogeneous clients [163]. The method of synchronous and asynchronous communication is also shown in the figure Figure 5. FL requires caution while scheduling procedures, particularly when working with IoT devices that have constrained computational power. Clients may use more computing resources as a result of frequent interactions and communications with the server, which will only make the system as a whole slower and more limited. The system can be sped up and consumption can be minimized by using optimal scheduling. New methods and algorithms are often proposed that would be able to schedule tasks correctly and do them more efficiently with the use of the least possible energy consumption and execution time, and at the same time with the execution required for processing. For example, Sun et al. [164], in their paper, discuss the optimal FL-based resource allocation, which they use in IoV. The authors focus on the Delay Energy Product (DEP), where they are able to derive the optimal balanced delay and energetic power consumption for scheduling. Their goal is to select the model with the lowest DEP value. Zhang et al. [165] also focus on MEC in a novel Federated Adaptive Training framework, which aims to adaptively assign an appropriate workload to each client. The algorithm adopts an adaptive workload allocation strategy and minimizes the difference when running the process in a heterogeneous device environment, as opposed to FedAvg, which uses a fixed number of iterations. Mahmood et al. [166] focus on minimizing task duration by optimally allocating resources such as local and edge computing capacities, transmission performance, and optimal task segmentation.

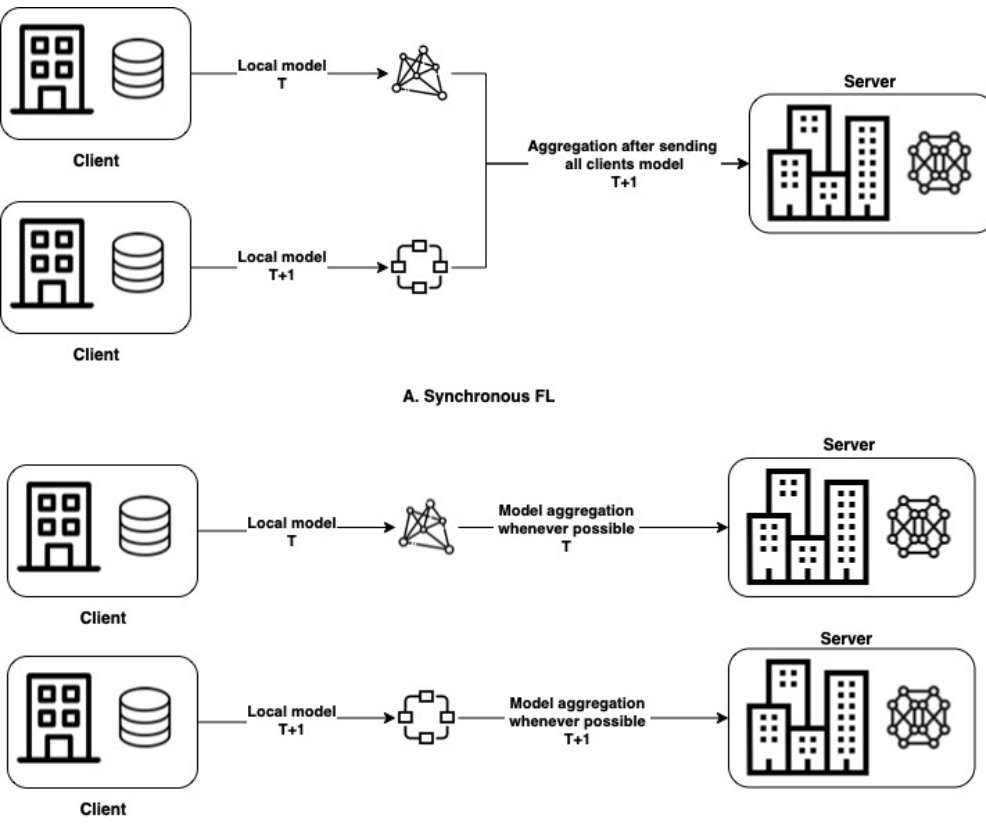

**Figure 5.** Synchronous and asynchronous FL communication.

### 6.8. Privacy Issue

However, due to limited resources, FL system privacy protection algorithms may not work on these devices. Therefore, in addition to providing strong privacy guarantees, new techniques must be developed that are communicationally efficient, computationally affordable, and capable of handling dropped participants. A number of studies focus on the problems related to peripheral devices whose computing power is insufficient. In these cases, it is impossible to use sophisticated encryption and communication security because it can hinder the computing capabilities of these devices. Wang et al. [167] propose a lightweight shared secret and weight mask privacy protection protocol based on a random secret sharing mask scheme. The advantage of this protocol is that it can, not only protect gradient privacy without losing model accuracy, but also resist device dropout attacks and device-to-device collusion. The Internet of Vehicles (IoV) can also be considered a paradigm placed at the edge of the network and viewed as a paradigm with limited computing power. The paper [168] proposes a privacy-aware IoV service deployment method with FL in cloud-edge computing. In this case, FL is ensured using distributed edge servers. These servers' task is to aggregate model weights, avoiding the transmission of flood data, and at the same time, these data are homophobically encrypted. Zhang et al. [169] solves the security of mobile devices and peripheral computing devices. They focus on cryptographic operations, which can often be computationally demanding. In the article, I propose a Verifiable Privacy-preserving Federated Learning scheme that prevents gradients from leaking during the transfer phase. At the same time, they also present an online/offline method for the implementation of signatures of light gradient integrity verification.

### 6.9. Scalability

The FL process must be highly scalable. In an IoT environment, many clients can participate in model training. In FL communication, it is also necessary to ensure good scalability of the whole system. With a large number of clients, it can happen that some clients lose connection to the server, for example, due to poor quality and weak signal or due to low battery as it can be in the case of mobile devices. It is necessary to ensure that such a system can deal with these situations. In this case, model training can be performed through efficient client selection [170]. The scalability and effectiveness of FL training and client selection have been the subject of several studies that have proposed different types of quality client selection. Clients who communicate wirelessly with a connection that is not very reliable can sometimes be problematic. In this case, only trustworthy clients are sought for training [171]. Ye et al. [172] developed FL framework based on a combination of models selected by clients for computing vehicle edges. Instead of randomly selecting FL clients, they use a client selection method based on contract theory. The scalability of FL systems is the subject of further studies. Zhang et al. [173] mainly solve scalability problems and low communication latency between the client and server. The authors propose an FL framework based on a cooperative, mobile edge network that can reduce latency compared to cloud communication. Distributed edge servers, which facilitate communication with other edge servers, manage the coordination of mobile devices in this case. Moreover, in the article [174] the authors describe and discuss the problems arising during local training of the model on the client side and the scalability of the server side. In this case, their goal is to reduce scalability problems, describe individual advantages and disadvantages, and propose the most usage scenarios.

### 6.10. Summary

Mobile and edge devices have constrained computing and storage capabilities. In particular, the battery life is limited, and the power consumption and latency when using such applications are high. In addition, many edge devices have various hardware and software specifications. For these computationally demanding applications, edge computing is a key technology that enables local model training at the network's edge. Several studies and experiments are focused on solving these subproblems.

### 7. Future Work

This article mainly focuses on the overview of FL applications and techniques in practical applications. FL, a recently developed distributed ML technique that can be regarded as a new research field that is constantly expanding and improving, is still a significant challenge. With FL and edge devices, various problems and issues must be resolved. For instance, one of them is how readily heterogeneous devices can communicate with one another. There are a variety of methods for reducing communication costs. Three types of these methods can be distinguished: model compression, decentralized training, and importance-based updating. Wang et al. [142] study an orthogonal approach that identifies irrelevant updates made by clients and excludes them from being sent to reduce the network and communication overhead. In this case, it is possible to reduce the communication overhead among devices and, simultaneously, ensure the model's convergence. Hiu-Po Wang et al. [175] aim to reduce computational and bidirectional communication overhead while maintaining the high performance of finite models. In addition, ML and AI techniques are now widely applied in wireless communication [176], as the computing power of devices has increased. Chen et al. [177] also proposed a collaborative learning framework by considering the impact of wireless factors on the participants in the FL scenario. Some of these methods adapted the compression strategies of the model. The drawbacks of this approach may be the degradation of model accuracy or high computational cost. Some trade-offs need to be made before communication is initiated, whereby several local training runs need to be performed to find the optimal number of iterations. The FL method can be made more scalable by using efficient optimization techniques that are empirically implemented and tested. The client must compromise its resources each time it communicates with the server. To deal with the constrained resources of clients, a resource optimization algorithm that considers such trade-offs is required. However, there is a lot of research that could theoretically be implemented into the FL system, which could improve communication between devices, increase data transfer, and better coverage [178], as well as nonorthogonal multiple access, which can significantly improve the spectral efficiency of these networks [179]. In this case, there is also research on communication effectiveness [180–183]. It is possible that future edge devices will use the 5G and 6G networks for communication, which will enable reliable networks for FL, while the true 6G will rely on two promising technologies, NOMA and reverse communication, which will ensure high data transfer speeds and robust connections [184]. Nowadays, privacy and security of communication are just as important as data transfer speed. Large-scale aforementioned 5G and 6G networks can handle the speed of communication and are gaining popularity. FL promises a high level of security, and private data must not be leaked or shared among devices. Given that most edge devices communicate using wireless networks, it is also necessary to focus on this type of communication and security. However, sharing private data makes the network vulnerable to data leakage and privacy. In this context, this paper [185] proposes an efficient and secure data sharing scheme using community segmentation and a blockchain-based framework for vehicular social networks. Similarly, blockchain technologies could be used not only for IoV but also for FL systems, providing a higher level of security [186].

Even though many research solutions have been proposed to lessen the challenges of implementing FL in the EC paradigm, problems still have not been solved. The EC paradigm can be applied to many research projects in FL. Some clients in the underlying IoT network environment may access more data than others in the FL-based IoT environment. Different participants train at noticeably different times due to the various memory sizes and availability, affecting the number of data samples [183]. This might result in significant discrepancies that make training and communication issues worse. More research is required to manage these differences within a local training dataset. The clients' mobility can significantly alter the system's general behavior. Before the training begins, there may be a lot of active clients on a network. However, after a while, most of them may stop using the network for various reasons, such as signal loss, battery life, etc. How to address the

mobility issues of IoT devices and guarantee a successful training of the federated model is thus a potential research area [187]. Getting FL clients to disclose details about their model could be another issue. It is essential to create a robust incentive mechanism in this circumstance. Given that FL participants may have limited resources or business rivals, it is also crucial to design a strategy that distributes the total profit to ensure participants' long-term engagement [188]. Additionally, more consideration needs to be given to thwarting the attacks of a rival who wants to take the majority of the incentives. In order to prevent hostile clients during training, the FL structure compels us to consider incorporating a trust model. On the other hand, in light of this, we must eliminate dishonest clients. The participating clients are anticipated to update their model parameters by the global model throughout the FL training process. This indicates that there is only one model in which each client can participate. However, clients might want to train multiple models [189]. Since local training and global model aggregation are separated, clients can employ various learning algorithms. For various purposes, it might be necessary, for instance, to use a federated approach to create multiple distinct models. It is critical to consider options in this situation that could enable the training of multiple models. As a result, it is important to analyze and use the right strategies. The accuracy of the global model could theoretically be increased by training more complex models on edge devices with heterogeneous hardware.

There are several application that use FL. Nowadays, edge devices are not only used for processing data from sensors and actuators but also for deploying models that can solve simple problems. Discussions arise about deploying artificial intelligence at the edge of the network, where these devices would participate in training models. In our case, we would like to focus on edge devices that process data from video cameras used in transport to monitor the traffic situation in smart cities and highway tolls. Our use case would focus on the hardware requirements of the individual frameworks, the edge devices benchmark designed for more complex tasks, and the efficiency of communication among devices. We can include more powerful microcomputers or AI accelerators among such devices. Various combinations of sensors are used to monitor traffic in cities. The aim is to replace traditional methods of sensing road traffic with expensive LiDAR sensors or ToF (Time of Flight) and replace them with cheaper camera-only systems. Each such system would have its sample of data from different cameras, angles, and times, which would be stored locally in edge devices that are placed close to the cameras. In this case, storing all the data on one central server is impossible because the data owners are different organizations. These organizations protect their data and do not disclose any details about it. Therefore, we decided to use the FL training model. Since this is a heterogeneous environment with different amounts of local data, our goal will also be to focus on the communication speed, efficiency, and hardware and software troubleshooting of these devices that will be used.

## 8. Conclusions

Federated learning is a promising approach for utilizing the ever-increasing computational power of the devices on the edge of the network and the large and diverse datasets to train machine learning models without compromising data privacy. Privacy is key for applications in healthcare or finance, as they are inherently extremely sensitive and sharing these data are often impossible. Before these architectures are widely used in commonplace applications, many research questions still need to be resolved, despite a few examples of FL being successfully used in production settings. FL has become a cutting-edge paradigm in response to the growing computational power of devices such as smartphones, wearables, and autonomous cars, as well as worries about the security of sensitive data. Due to the increased need for local data storage and the relocation of ML computations to end devices while minimizing data transfer overhead, researchers have tried to implement FL architectures in various domains. An overview of the FL paradigm, which is gaining popularity, is provided in this article. We focused on the most recent materials and publications, as we discussed FL's fundamental architecture, communication, design, and analysis. We discussed the basic requirements that FL must meet, the difficulties involved, the potential

for deployment and use in practical applications, and the frameworks with which FL is compatible. We have focused on the overview of the frameworks and examined the operating systems they supported and the potential for deployment on various edge devices. We discussed potential future directions for FL use in diverse IoT environments. We also considered the problems and difficulties that must be solved for edge devices that are resource-constrained and have limited computational power, even though the hardware specifications they employ are not always listed in the literature.

**Author Contributions:** Conceptualization and methodology, A.B. and I.Z.; formal analysis and supervision, E.K. and I.Z.; writing—original draft preparation, A.B.; writing—review and editing, A.B., E.K., J.K. and I.Z.; funding acquisition and project administration, J.K. and I.Z. All authors have read and agreed to the published version of the manuscript.

**Funding:** This publication is the result of the APVV grant ENISaC - Edge-eNabled Intelligent Sensing and Computing (APVV-20-0247). This work has received funding from the European Union's Horizon 2020 Research and Innovation Programme under grant agreement N°856670.

**Institutional Review Board Statement:** Not applicable.

**Informed Consent Statement:** Not applicable.

**Data Availability Statement:** Not applicable.

**Conflicts of Interest:** The authors declare no conflict of interest. The funders had no role in the design of the study; in the collection, analyses, or interpretation of data; in the writing of the manuscript; or in the decision to publish the results.

## Abbreviations

The following abbreviations are used in this manuscript:

| | |
|---|---|
| FL | Federated Learning |
| IoT | Internet of Things |
| IIoT | Industrial IoT |
| IoV | Internet of Vehicle |
| AI | Artificial Intelligence |
| ML | Machine Learning |
| FedAvg | Federated Averaging |
| DT | Decision Tree |
| RF | Random Forest |
| SVM | Support Vector Machine |
| KNN | K-nearest Neighbour |
| P2P | Peer-to-Peer |
| GDPR | General Data Protection Regulation |
| CCPA | California Consumer Privacy Act |
| PDPA | Personal Data Protection |
| FATE | Federated AI Technology Enabler |
| MPC | Multi-party Computation |
| OpenFL | Open Federated Learning |
| CA | Certificate Authority |
| TFF | TensorFlow Federated |
| FC | Federated Core |
| SDK | Software Development Kit |
| MEC | Mobile Edge Computing |
| DEP | Delay Energy Product |
| SBC | Single Board Computer |
| FPGA | Field Programmable Gate Array |

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
