# Peer review of "Federated Learning for Edge Computing: A Survey"

_applsci, doi:10.3390/app12189124_

Round 1

Author Response

Dear revievew 1,

After incorporating all the reviewers' comments, we submit a corrected version of our research paper entitled "Federated Learning for Edge Computing: A Survey". Thank you for your time and positive feedback on our review article. We have tried to incorporate the comments you gave us, so hopefully, it will be fine.

Answers to your questions:
- The article was just an overview of FL technologies and methods that are currently in use. In this case, we have not done any experiments yet, but we plan to do so in a future research.
- In the review, we also discussed FL on mobile devices and applications created using these devices. These applications are covered in Section 5.1 (in the revised version of the review article). For future 5G and 6G network technologies, we believe that bandwidth will not need to be addressed in this case, as these technologies achieve very decent speed results.
- We think that privacy and security are one of the most significant advantages of FL. We incorporated these comments in chapter 3.5 Privacy and security, and chapter 6.8 Privacy issue. Both chapters were supplemented with citations and accompanying texts.
- This survey focused on edge devices and FL deployment opportunities at the network's edge. FL is a relatively new and popular topic that many researchers are exploring. In our case, we focused on the constraints and challenges that must be addressed when deploying FL on edge devices. Certain limitations in terms of computational power, memory, and communication capabilities can affect the accuracy of the global model. At the same time, many devices with local data can contribute to a more accurate model because we have a larger amount of data to train.
- The advantages of using FL compared to traditional methods are summarized in chapter 3.7 Summary. The most significant advantage is that it can train a global model without sharing private data to data centers and clouds. Only aggregated local models from several devices are sent simultaneously, e.g., to the central server where the global model is created.

To simplify the reviewers' work, we have included the Differences.pdf file as unpublished material and our responses as an appendix, which show all the differences between the original and revised versions of our manuscript. 

Thank you for your review of this manuscript. 

Sincerely, 

Alexander Brecko

Reviewer 2 Report

This paper provides an overview of the methods used in FL with a focus on edge devices with limited computational resources. Paper also present FL frameworks that are currently popular and that provide communication between clients and servers. In this context, various topics are described, which include contributions and trends in the literature. This includes basic models and designs of system architecture, possibilities of application in practice, privacy and security, and resource management. Challenges related to the computational requirements of edge devices, such as hardware heterogeneity, communication overload or limited resources of devices, are discussed.  I have the following comments:

1) This paper's contribution is unclear; the authors need more explanation about the contribution of this work. 

2) They only discuss the basics of federated learning, which is not enough. There exist a lot of work that discusses the basics of federated learning.

3) The authors should also discuss recent literature on federated learning in edge computing in detail.

4) A good review paper consists of informative figures, tables, etc. However, there are only have few figures and a table. Paper lacking it.

5) What is the difference between this work and the work "Federated learning and next generation wireless communications: a survey on bidirectional relationship" and "RL/DRL meets vehicular task offloading using edge and vehicular cloudlet: a survey"? Explain it in the revised paper.

6) Paper also needs proper proofreading as there to exist several typos and grammar errors.

7) It is recommended to study all survey/review papers on federated learning/edge computing and compare them with this paper. Also, make a table for a detailed comparison. 

8) I would like and expect to see the revised version of this paper.

Author Response

Dear Reviewer 2,

After incorporating all the reviewers' comments, we submit a revised version of our research paper titled "Federated Learning for Edge Computing: A Survey." Thank you for your time and feedback on our review article. We've tried to incorporate the comments you've given us, so I hope it's okay.

Answers to your questions:
- In this paper, we have tried to introduce the basic characteristics of FL and describe the challenges and problems associated with FL peripherals and methods in the following sections. We've edited and added a few paragraphs to define them more clearly. We tried to enrich the work with problems and challenges on peripheral devices. We focused primarily on ARM-based devices because we believe this architecture will continue to expand and improve at the network's edge. So far, we have found few references and resources that speak directly about supporting this architecture.
- We included pictures and tables that were missing from the article and worked on grammatical errors and typos
- We tried to describe the contributions and incorporate them into the work. We hope we managed it according to your expectations.

To make the reviewers' work more accessible, we have included Differences.pdf as unpublished material and our responses as an appendix, all differences between the original and revised versions of our manuscript.

I appreciate your attention to this manuscript.

Sincerely

Alexander Brecko

Reviewer 3 Report

The authors provided an overview of the approaches used in FL with a focus on edge devices with limited computational resources. The idea is interesting. However, I have the following concerns. 

1. A check on grammatical issues is required. 

2. Recent FL-related research is missing. Some are mentioned as follows.

-> "FBI: A Federated Learning-Based Blockchain-Embedded Data Accumulation Scheme Using Drones for Internet of Things," in IEEE Wireless Communications Letters, vol. 11, no. 5, pp. 972-976, May 2022, doi: 10.1109/LWC.2022.3151873.

-> "Recent Advances on Federated Learning for Cybersecurity and Cybersecurity for Federated Learning for Internet of Things," in IEEE Internet of Things Journal, vol. 9, no. 11, pp. 8229-8249, 1 June1, 2022, doi: 10.1109/JIOT.2022.3150363.

3. FL on edge is a very popular topic. Highlight the novelty based on the limitation of existing works. 

4. Add a summary at the end of each section.

5. A discussion on scalability is required. 

6. A discussion on security challenges and strugglers regarding FL would be better. 

Author Response

Dear Reviewer 3

After incorporating all the reviewers' comments, we present the updated version of our research paper entitled "Federated Learning for Edge Computing: A Survey". Thank you for taking the time to review and consider our survey document. We have tried to incorporate the comments you have provided, so we hope this will be alright.

Responses to your questions:
- We have worked on the scalability and security topics in the paper. We have tried to support all of these changes with the latest references and sources.
- We have attempted to correct grammatical and typographical errors in the article.
- We have added a Summary section to each section.

To make the reviewers' job more accessible, we have included the Differences.pdf file as unpublished material and our responses as an appendix, in which all the differences between our manuscript's original and revised versions are color-coded. 

I appreciate your consideration of this manuscript. 

Sincerely, 

Alexander Brecko

Reviewer 4 Report

This article is a comprehensive review of federated learning focusing on edge computing. The authors summarized published federated learning methods on edge devices as well as widely-used frameworks. The authors also point out the current challenges in federated learning and edge computing, mostly around constrained computing resources and system heterogeneities. I would recommend to publish the review but with minor change. 

As the author pointed out, one of the challenges is the "heterogeneous hardware", however, the author overlooked another source of heterogeneity, data heterogeneity. Non-identically and non-independent (non-IID) distribution of the data is a well known challenge in federated learning. For example, there are many studies demonstrating FedAvg is not robust to non-IID data and many modifications on FedAvg (SCAFFOLD, FedDyn, etc.) have been published to mitigate this problem and in the meantime reduce the communication rounds for convergence (which also reduce communication cost). It is worth adding this discussion to the review. 

Author Response

Dear Reviewer 4

After incorporating all the reviewers' comments, we present the corrected version of our research paper entitled "Federated Learning for Edge Computing: A Survey". Thank you for taking the time to review and positive consider our survey document. We have tried to incorporate the comments you have provided, so we hope this will be alright.

Responses to your questions:
- We have incorporated your comment and added a chapter to the article. We have included the primary and most important information that we felt was essential. All of this information is summarized in Section 6.6 Heterogeneous Data.

To make the reviewers' work more accessible, we have included the Differences.pdf file as unpublished material and our answers as an appendix, with all the differences between our manuscript's original and revised versions. 

I appreciate your consideration of this manuscript. 

Sincerely, 

Alexander Brecko

Round 2

Reviewer 2 Report

1) Thank you so much for addressing my comments. I think future research works and directions still need to be improved. It suggested discussing how the new technologies can be efficiently integrated into Federated learning-enabled Edge computing. These technologies include but are not limited as; intelligent reflecting surfaces "Opportunities for physical layer security in UAV communication enhanced with intelligent reflective surfaces"; non-orthogonal multiple access which can improve the spectral efficiency of these networks significantly "Joint spectrum and energy optimization of NOMA-enabled small-cell networks with QoS guarantee"; millimeter wave/Terahertz communication which is considered to be the new spectrum opportunity in next-generation systems "A survey on terahertz communications"; security is another important aspect which need to be studied in future "A secure data sharing scheme in Community Segmented Vehicular Social Networks for 6G"; backscatter communication for battery-free communication "NOMA-enabled backscatter communications for green transportation in automotive-Industry 5.0".

2) Based on the above observations, I strongly recommend revising the future work section for the reader of this paper. 

3) This paper covers all the recent advances? If not, it is better to report all the technical works related to the survey. I found this work, "Optimal resource allocation and task segmentation in iot enabled mobile edge cloud" please study and report along the all other literature.

4) Lot but not most minor, I again suggest proofreading this paper before accepting it.

Author Response

Dear Reviewer 2,

Thank you for your time on the manuscript and your positive feedback on the "Federated Learning for Edge Computing: A Survey" paper.

After incorporating all the comments from the 2nd round of reviews, we present a revised version of our research paper, so we hope this will be alright.

Response to comments:
- In the survey paper, we have reworked the "Future work" and "Conclusion" sections.
- We have added additional resources and references linked to our survey paper.

We have accepted all your comments and thank you for the inspiration for further research. To make the reviewers' work more accessible, we have included Differences.pdf as unpublished material. All differences between our manuscript's original and revised versions are marked in this appendix.

We appreciate your attention to the manuscript.

Sincerely

Alexander Brecko

Reviewer 3 Report

I am recommending to accept this paper.

Author Response

Dear Reviewer,

Thank you for your time on the manuscript and your positive feedback on the "Federated Learning for Edge Computing: A Survey" paper.

After incorporating all the comments from the 2nd round of reviews, we present a revised version of our research paper, so we hope this will be alright.

We have accepted all comments from other reviewers. To make the reviewers' work more accessible, we have included Differences.pdf as unpublished material. All differences between our manuscript's original and revised versions are marked in this appendix.

We appreciate your attention to the manuscript.

Sincerely

Alexander Brecko
